# Nickel-catalyzed acylzincation of allenes with organozincs and CO

Xianqing Wu [1], Chenglong Wang[1], Ning Liu[1], Jingping Qu[1] & Yifeng Chen [1] ✉

Transition metal-catalyzed carbonylative reaction with CO gas are among the central task in organic synthesis, enabling the construction of highly valuable carbonyl compound. Here, we show an earth-abundant nickel-catalyzed three-component tandem acylzincation/cyclization sequence of allene and alkylzinc reagent with 1 atm of CO under mild conditions. This protocol is featured by broad functional group tolerance with high reaction selectivity, providing a rapid and convenient synthetic method for the construction of diverse fully substituted benzotropone derivatives. Mechanistic studies reveal that the installation of a cyano group tethered to allene moiety enables the high regio- and stereoselectivity of this acylzincation of allene, allowing the selective formation of three consecutive C-C bonds in a highly efficient manner.

Transition metal-catalyzed carbometallation reaction of unsaturated hydrocarbons that simultaneously manufacture two C–C bonds provide expedient access to complex molecules from widely available feedstocks (Fig. 1a)[1,2]. Nevertheless, the well-established aforementioned reactions mainly leverage the electronically rich organometallics such as organolithium[3,4], Grignard[5–8] and organozinc[9–14] reagents as nucleophilic components (Fig. 1a, left). In contrast, the utilization of an acyl synthon as a nucleophile engaged in transition metal-catalyzed carbometallation of unsaturated carbon-carbon bonds is still largely underdeveloped, probably due to the lack of a practical synthetic route for the preparation of stoichiometric acyl metallic reagents from the corresponding acyl halide precursors[15,16], as well as the difficulty for the electron-deficient acyl metal species to undergo intermolecular migratory insertion towards unsaturated hydrocarbons (Fig. 1a, right)[17].

Until recently, our group reported the first nickel-catalyzed acyl-metallation of activated unsaturated hydrocarbons with the functionalized organozinc and atmospheric CO gas through the in-situ formation of catalytic reactive acyl nickel intermediate[18], in which the combination of nickel catalyst and bidentate or tridentate nitrogen-based ligand inhibits the formation of unreactive $Ni(CO)_4$ species[19]. This chemo-, regio- and stereoselective acylzincation protocol opens up new vistas in the arsenal of carbometallation reaction. Intriguingly, a cascade acylzincation/desulfonylative Smiles rearrangement process of benzenesulfonamide-substituted ynamide furnished the diverse tetrasubstituted substituted β-amino enones (Fig. 1b, i). While acyl-zincation of α,β-unsaturated ketones could initiate intramolecular aldol condensation reaction to provide highly substituted cyclopentenones (Fig. 1b, ii). Despite these preliminary advances, the further exploitation of the horizon of the acylmetallation triggered cascade reaction is highly desirable. In this context, the allene skeleton is broadly recognized as a reactive unsaturated carbon-carbon bond to participate in various carbometallation reactions, thus increasing the challenge to incorporate CO gas to accomplish the acylmetallation of allenes (Fig. 1b, iii)[20–26]. Additionally, the regulation of regio- and stereoselectivity is another obstacle to overcome.

Recently, the employment of cyclization of allene has emerged as a synthetically useful toolbox for the construction of polycyclic skeletons or heterocycles[27–38]. We envision that introducing a bifunctional cyano group in allene moiety may accelerate the acylmetallation process, as well as serve as a highly functionalized group to participate in tandem transformations. However, in addition to the aforementioned direct carbometallation of allenes, as well as the potential regio- and stereoselectivity of acylmetallation, the strong nucleophilic organometallic reagent might undergo 1,2-addition towards the nitriles to generate the imine[39]. Based on our continuous research interest in nickel-catalyzed carbonylation transformations[40–65], as well as the aim for developing atom-, step-economical and greener synthetic routes to access challenging scaffolds, herein,

[1]Key Laboratory for Advanced Materials and Joint International Research Laboratory of Precision Chemistry and Molecular Engineering, Feringa Nobel Prize Scientist Joint Research Center, Frontiers Science Center for Materiobiology and Dynamic Chemistry, School of Chemistry and Molecular Engineering, East China University of Science and Technology 130 Meilong Road, Shanghai, China. ✉e-mail: yifengchen@ecust.edu.cn

we report our latest finding on the nickel-catalyzed acylzincation initiated cascade cyclization reaction of cyano-substituted allenes with alkyl organozinc reagents under 1 atm of CO, affording various fully substituted benzotropones that frequently occurs in various natural products with biological interest (Fig. 1c)[66,67].

## Results and discussion

### Reaction optimization for substituted benzotropones synthesis

We commenced our studies by investigating the Ni-catalyzed acylzincation of allene **1a** with the $^n$BuZnCl **2a** under 1 atm CO as depicted in Fig. 2. The combination of Ni(acac)$_2$ and terpyridine-

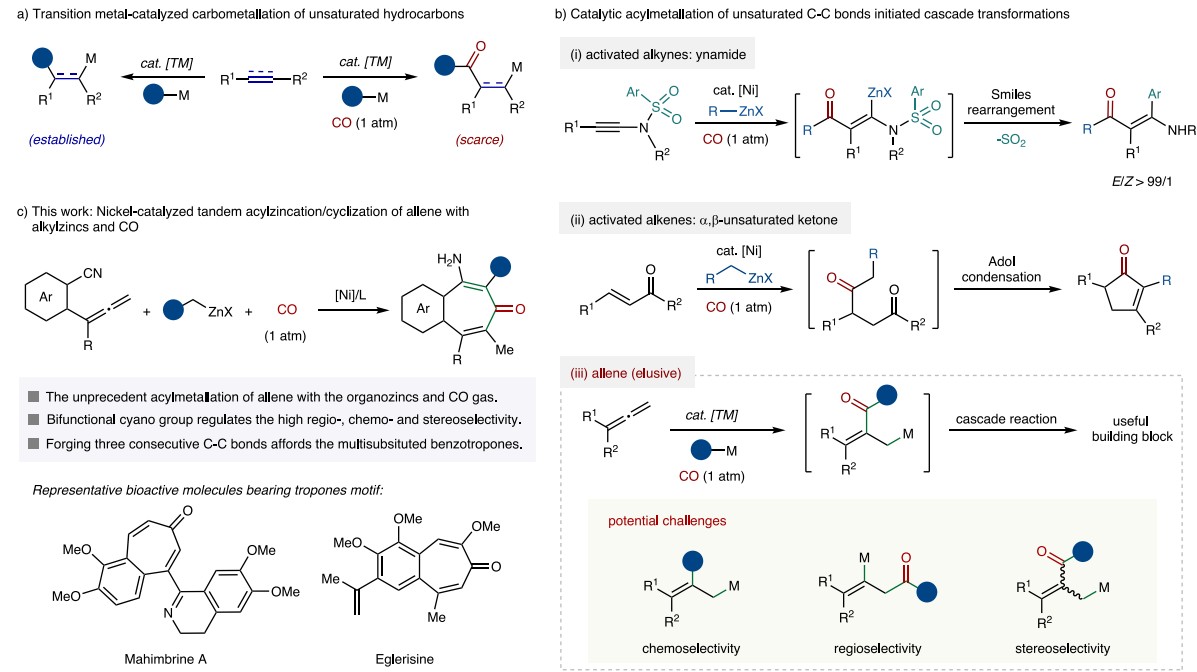

**Fig. 1 | Transition metal-catalyzed acylzincation of unsaturated hydrocarbons.** **a** Transition metal-catalyzed carbometallation of unsaturated hydrocarbons. **b** Catalytic acylmetallation of unsaturated C−C bonds-initiated cascade transformations. **c** Nickel-catalyzed tandem acylzincation/cyclization of allene with alkylzincs and CO. TM, transition metal; M, metal; X, halogen.

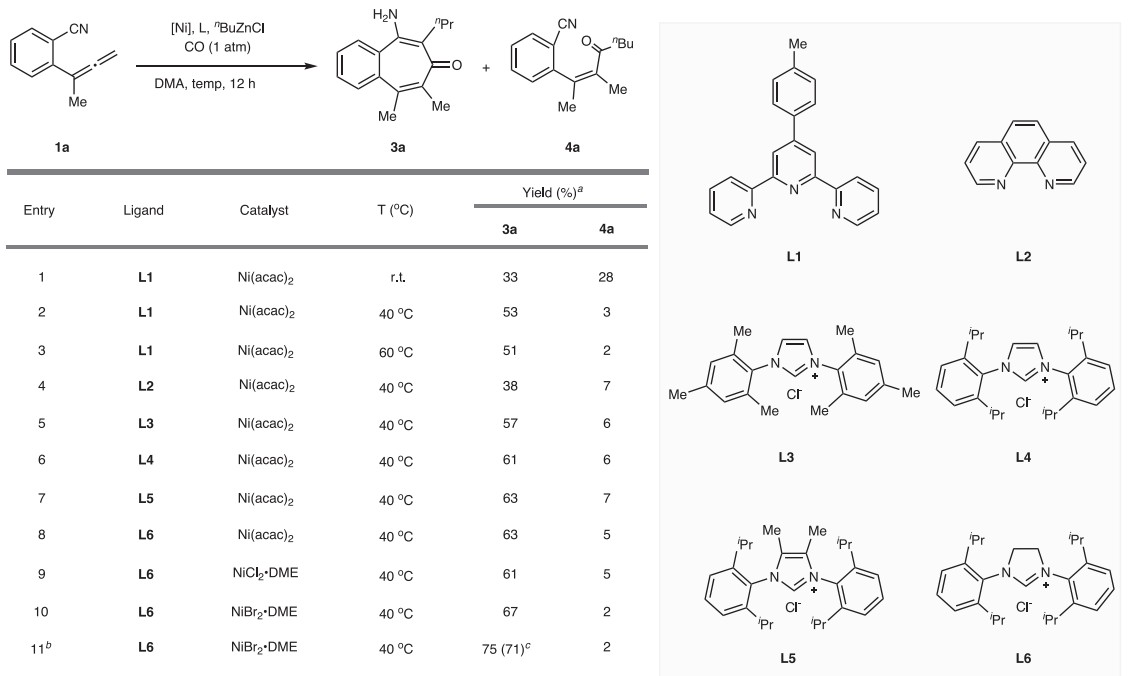

| Entry | Ligand | Catalyst | T (°C) | Yield (%)[a] | |
|---|---|---|---|---|---|
| | | | | **3a** | **4a** |
| 1 | **L1** | Ni(acac)$_2$ | r.t. | 33 | 28 |
| 2 | **L1** | Ni(acac)$_2$ | 40 °C | 53 | 3 |
| 3 | **L1** | Ni(acac)$_2$ | 60 °C | 51 | 2 |
| 4 | **L2** | Ni(acac)$_2$ | 40 °C | 38 | 7 |
| 5 | **L3** | Ni(acac)$_2$ | 40 °C | 57 | 6 |
| 6 | **L4** | Ni(acac)$_2$ | 40 °C | 61 | 6 |
| 7 | **L5** | Ni(acac)$_2$ | 40 °C | 63 | 7 |
| 8 | **L6** | Ni(acac)$_2$ | 40 °C | 63 | 5 |
| 9 | **L6** | NiCl$_2$·DME | 40 °C | 61 | 5 |
| 10 | **L6** | NiBr$_2$·DME | 40 °C | 67 | 2 |
| 11[b] | **L6** | NiBr$_2$·DME | 40 °C | 75 (71)[c] | 2 |

**Fig. 2 | Optimization of the reaction conditions.** The reaction was performed with **1a** (0.1 mmol), $^n$BuZnCl (**2a**) (0.15 mmol, 1.5 equiv), **CO** (1 atm), [Ni] (0.01 mmol, 10 mol%) and **L** (0.02 mmol, 20 mol%) in DMA (0.1 M) for 12 h. [a]Yield determined by $^1$H NMR using CH$_2$Br$_2$ as the internal standard. [b]NiBr$_2$·DME (0.0025 mmol, 2.5 mol%), **L6** (0.005 mmol, 5 mol%). [c]Isolated yield in the parentheses. DMA, N,N-dimethylacetamide; acac, acetylacetonyl; DME, 1,2-Dimethoxyethane.

based ligand **L1** could smoothly catalyze this acylzincation reaction in DMA at room temperature, delivering the desired product **3a** and uncyclized unit **4a** in 33% and 28% yield, respectively (entry 1). Raising the temperature to 40 °C could increase the reaction yield of product **3a** to 53% yield with simultaneous consumption of most of **4a** (entry 2), while continuing to elevate the temperature to 60 °C would have no effect on the outcome (entry 3). Further evaluating the ligand effect indicated that the bidentate 1,10-phenanthroline **L2** led to reduced yield (entry 4), while electron-rich NHC ligand **L3** could make the nickel center more electronically rich, thus accelerate the migratory insertion and improve the reaction efficiency, affording the product **3a** in 57% yield (entry 5)[68–70]. The subsequent NHC ligand screening revealed that the bulkier SIPr·HCl **L6** could yield the best result (entries 6–8). With respect to the catalyst, NiBr$_2$·DME showed a superior effect over NiCl$_2$·DME and Ni(acac)$_2$, furnishing product **3a** with 67% yield (entries 9–10). Intriguingly, decreasing the loading of both catalyst and ligand resulted in a higher yield, providing the benzotropone **3a** with 75% $^1$H NMR yield (71% isolated yield) (entry 11).

## Substrate scope

With the optimized conditions in hand, we next explored the substrate scope of alkyl zinc reagents (Fig. 3). Simple alkyl groups including *n*-butyl (**3a**), phenylpropyl (**3b**), *n*-decyl (**3c**), ethyl (**3d**) and isobutyl (**3e**) could be successfully incorporated into the products in moderate yields. The alkene functionalities including both terminal and internal alkenes were well compatible with this acylzincation protocol, affording the corresponding products **3f** and **3g** in good yields. Additionally, various functionalized alkylzinc reagents, including those possessing halogens such as Cl

(**3h**) and F (**3i**), ether (**3j**), ester (**3k**), and cyano group (**3l**) were smoothly converted to the corresponding benzotropone products. The protocol was not suitable with the utilization of methyl and benzyl zinc nucleophiles.

Next, we turned our attention to investigating the generality of this acylzincation protocol towards the allene substrates (Fig. 4). The electron-neutral substitution pattern on the aromatic ring played little influence on the reaction efficiency, giving the corresponding products in 57–63% yields (**3m**–**3p**). Besides, electron-donating substituents on the aromatic rings were also compatible, affording the desired products **3q**–**3s** in moderate to good yields. In addition, halogen including chlorine and fluorine-incorporated phenyl allenes were also suitable substrates to provide the corresponding desired products **3t**–**3v**, albeit in lower yields. The effects of substituents tethered to the allene moiety were also investigated. It was found that allenes bearing other alkyl substituents such as ethyl and *n*-butyl groups could also proceed smoothly with this protocol, affording products **3w** and **3x** in moderate yields.

## Synthetic utility

To demonstrate the synthetic utility of this protocol, we performed the following transformations (Fig. 5). Firstly, a 1.0-mmol scale reaction was carried out to afford the benzotropone **3a** in a higher 76% isolated yield (Fig. 5a). Furthermore, the fully substituted benzotropone products could readily undergo various derivatizations to access diverse structural motifs (Fig. 5b). The benzotropone **3a** generated from allene **1a** under standard conditions could be hydrolyzed by in-situ workup process with 3 M HCl to afford the 1,3-cycloheptandione product **5** (Fig. 5b, i). The carbonyl shift reaction of benzotropone could be smoothly

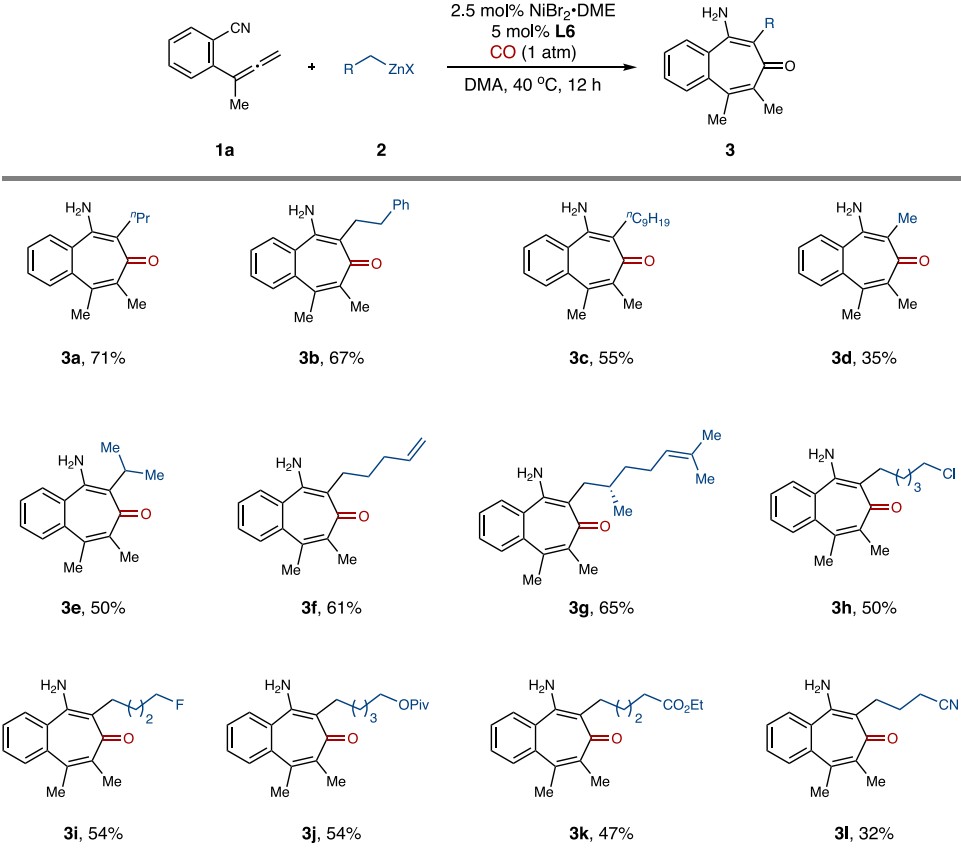

**Fig. 3 | Substrate scope of alkylzincs.** The reaction was performed with **1a** (1.0 equiv), **2** (1.5 equiv), **CO** (1 atm), NiBr$_2$·DME (2.5 mol%), **L6** (5 mol%) in DMA (0.1 M) at 40 °C for 12 h. DMA, *N,N*-dimethylacetamide; DME, 1,2-Dimethoxyethane; X, halogen.

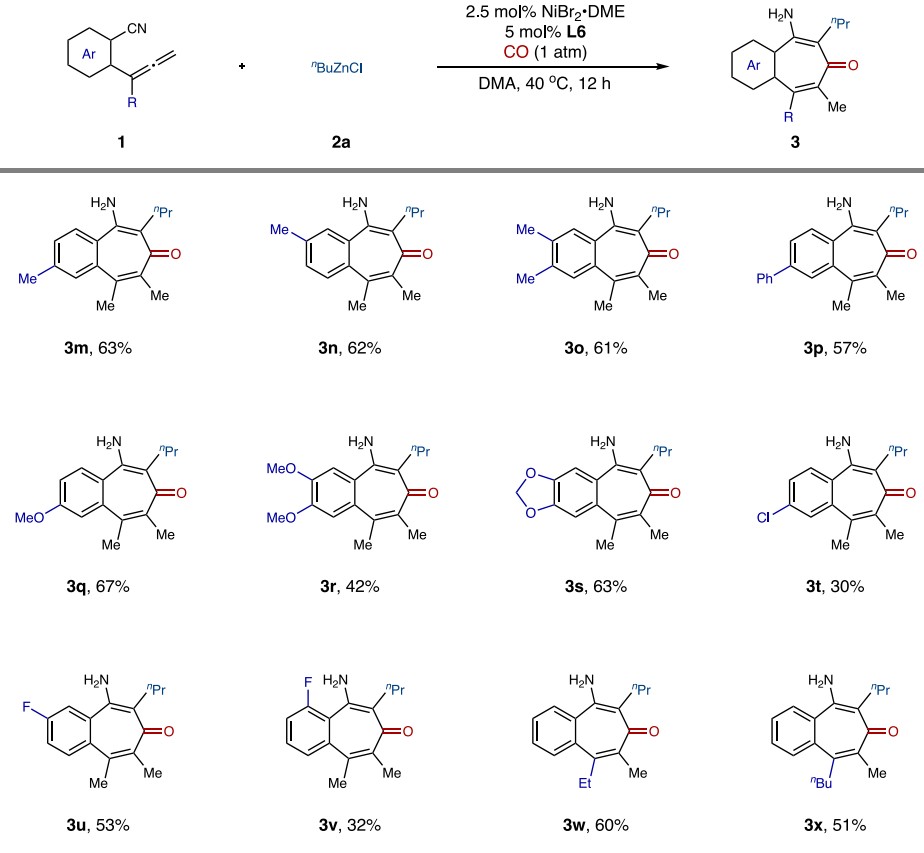

**Fig. 4 | Substrate scope of allenes.** The reaction was performed with **1** (1.0 equiv), **2a** (1.5 equiv), **CO** (1 atm), NiBr₂•DME (2.5 mol%), **L6** (5 mol%) in DMA (0.1 M) at 40 °C for 12 h. DMA, *N,N*-dimethylacetamide; DME, 1,2-Dimethoxyethane.

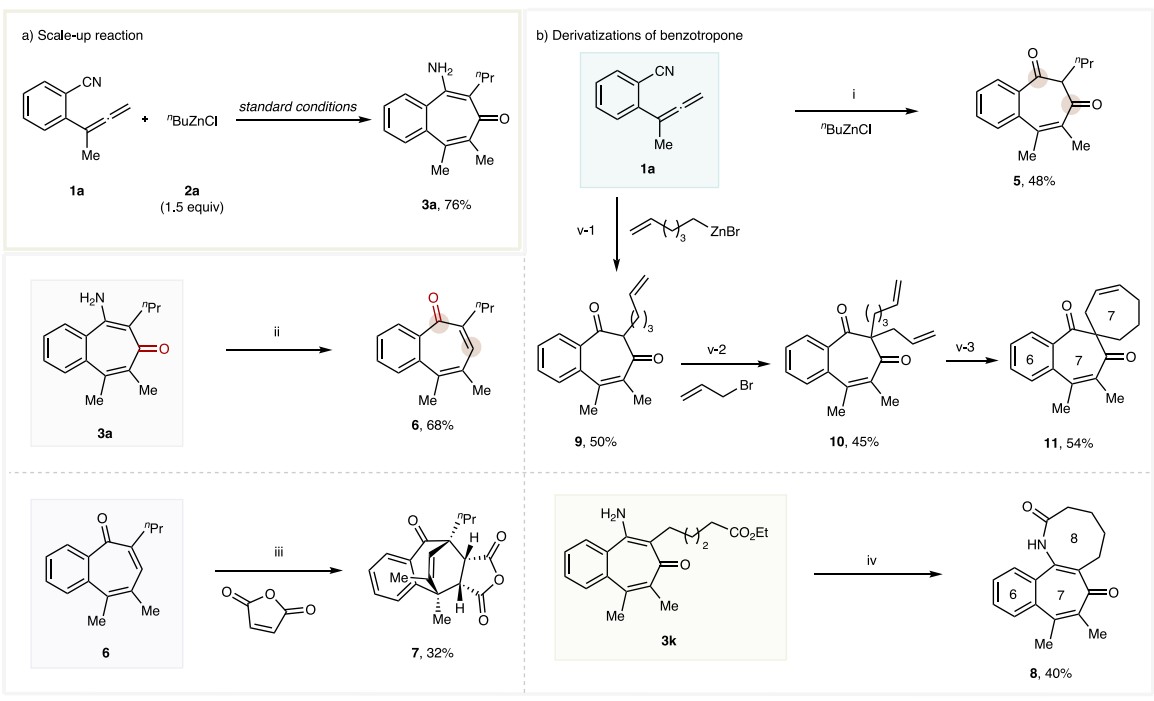

**Fig. 5 | Synthetic application. a** Scale-up reaction. **b** Derivatizations of benzotropone. Reaction conditions: (i) Standard conditions; then 3 M HCl, THF, 70 °C, 12 h. (ii) DIBAL-H, DCM, 0 °C, 0.5 h; then 3 M HCl, THF, 70 °C, 2 h. (iii) Toluene, reflux, 18 h; (iv) NaH, DMSO, 80 °C. (v) (1): Standard conditions; then 3 M HCl, THF, 70 °C, 12 h. (2): EtONa, EtOH, r.t. (3): 15 mol% Grubbs II catalyst, DCM, 40 °C. THF tetrahydrofuran, DMSO Dimethyl sulfoxide, DCM Dichloromethane.

obtained upon the reduction of **3a** with DIBAL-H followed by in-situ hydrolyzation with 3 M HCl, delivering the carbonyl transposition product **6** (Fig. 5b, ii), which can be further transformed to endocyclic product **7** through Diels-Alder reaction with maleic anhydride (Fig. 5b, iii). The benzotropone product **3k** bearing an ester functional group could undergo intramolecular transesterification to give the benzo-7,8-bicyclic product **8** (Fig. 5b, iv). The benzotropone **3f** generated from allene **1a** with alkenyl substituted alkylzinc reagent under standard conditions could be hydrolyzed by in-situ workup procedure with 3 M HCl to afford the 1,3-cycloheptandione product **9**, which next could be transformed into **10** through α-allylation of 1,3-diketone. Finally, the medium-sized spiro product **11** could be obtained by intramolecular RCM reaction (Fig. 5b, v).

## Mechanistic investigation

To gain more insight into the reaction process, we conducted several experiments to validate the mechanism (Fig. 6). First, manipulation for investigating the effect of the cyano group was performed. When allene **1n** without cyano group was employed as a substrate, acylzincation product **4n** was obtained in 22% yield ($E/Z = 6/1$) with recovery of **1n** (30% yield), which indicated that the cyano group not only served as an electrophilic component and significantly promoted the reaction efficiency, but also acted as a directing group to improve the stereo- and regioselectivity of this reaction (Fig. 6a). In addition, we also performed the real-time tracing experiment of this acylzincation/cyclization sequence (Fig. 6b). Only a small amount of **4a** was observed within the first 30 minutes of the reaction, and no product could be observed. With prolonging the reaction time, the concentration of benzotropone **3a** began to increase with the consumption of both **1a** and **4a**. The starting material **1a** was completely

consumed within 5 h, while the remaining **4a** was fully transformed into product **3a** in about 10 h. The above results indicated that compound **4a** might be an intermediate of this reaction. To verify the required parameters for converting intermediate **4a** to the cyclized product **3a**, we carried out the following control experiments. When **4a** was subjected to the standard conditions without the addition of organozinc reagent, no benzotropone product **3a** was observed. While **3a** could be produced in 72% yield with the omission of catalyst and ligand (Fig. 6c). These combined results demonstrated that organozinc reagent was necessitated for the cyclization process of intermediate **4a**. On the basis of the above preliminary results, a plausible mechanism for this protocol is proposed (Fig. 6d). Transmetallation of the nickel catalyst with alkylzinc reagent generates alkyl nickel species **A**, which next undergoes 1,1-insertion into CO to provide acyl nickel intermediate **B**. Subsequent migratory insertion towards intermolecular allene leads to allylnickel species **C**, which undergoes transmetallation with another alkylzinc reagent to furnish the organozinc intermediate **D** with the regeneration of reactive catalyst species **A**. Protonation followed by intramolecular condensation sequence yields the final benzotropone product **3**.

In conclusion, we have developed a nickel-catalyzed tandem acylzincation/cyclization sequence of allenes and organozinc reagents under atmospheric CO pressure, providing expedient access to various fully substituted benzotropone derivatives. The incorporation of a cyano group that serves as an electrophilic component in the reaction is crucial for regulating the stereo- and regioselectivity of the acylzincation step, enabling the highly selective formation of three consecutive C-C bonds. Further efforts into nickel-catalyzed asymmetric acylzincation of unsaturated hydrocarbons are ongoing in our laboratory.

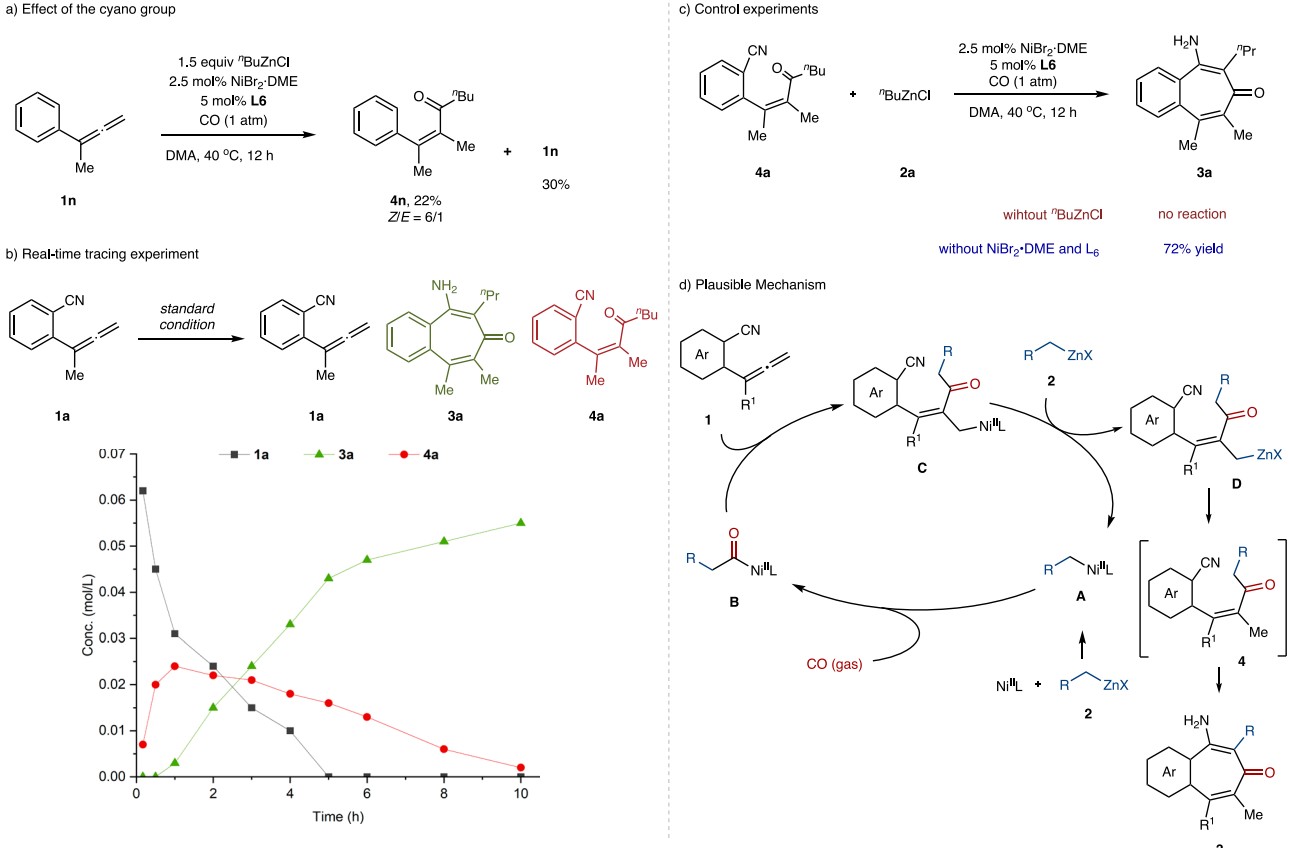

**Fig. 6 | Mechanistic experiment. a** Effect of the cyano group. **b** Real-time tracing experiment. **c** Control experiments. **d** Plausible mechanism.

## Methods

### General procedure for the acylzincation of allenes

A 10-mL oven-dried tube charged with NiBr$_2$·DME (2.5 mol%) and **L6** (5 mol%) was evacuated and backfilled with N$_2$ three times. The reaction mixture was evacuated again and backfilled with CO (1 atm, balloon), followed by the addition of DMA (0.1 M), allene (1.0 equiv) and alkylzinc reagent (1.5 equiv) at r.t. The tube was screw-capped and the reaction mixture was allowed to stir at 40 °C oil bath for 12 h. The mixture was quenched with saturated aqueous NH$_4$Cl and extracted with EtOAc. The separated organic layer was washed with brine, dried over anhydrous Na$_2$SO$_4$, and concentrated under reduced pressure to yield the crude product, which was purified by silica gel flash column chromatography. to afford products **3**.

## Data availability

All data to support the conclusions are available in the main text or the Supplementary Information. All other data are available from the corresponding author upon request.

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

## Acknowledgements

This work was sponsored by the National Natural Science Foundation of China (22171079, 22371071), Natural Science Foundation of Shanghai (21ZR1480400), Shanghai Rising-Star Program (20QA1402300), Shanghai Sailing Program (23YF1408800), Shanghai Municipal Science and Technology Major Project (Grant No. 2018SHZDZX03), the Program of Introducing Talents of Discipline to Universities (B16017), the Fundamental Research Funds for the Central Universities and the China Postdoctoral Science Foundation (2021M701197, 2023T160215). We thank Research Center of Analysis and Test of East China University of Science and Technology for the help on NMR analysis.

## Author contributions

Y.C. conceived the projects. X.W., C.W., N.L. performed the experiments under the supervision of J.Q. and Y.C. X.W. and Y.C. wrote the manuscript with the feedback of all other authors.

## Competing interests

The authors declare no competing interests.
