## [Peer Review File · Nature Communications]

Nickel-Catalyzed Acylzincation of Allenes with Organozincs and COReviewers' Comments:

Reviewer #1:

Remarks to the Author:

Chen and coworkers have made an exciting advancement in the realm of transition metal-catalyzed carbomatalation chemistry. Specifically, they have successfully achieved the nickel-catalyzed acylzincation of allenes using organozincs and CO, providing an efficient method for constructing fully substituted benzotropones, which are commonly found in various bioactive natural products. The same research group previously reported on the seminal work to probe the feasibility of acylmetallation of activated unsaturated hydrocarbons earlier this year (*Nat. Synth.* 2023, 2, 261).

Comparing with the reported polarized unsaturated hydrocarbons such as ynamide and enones by the same group, the allene functional group presents a unique challenge due to its three potential reaction sites. Achieving the acylmetallation reaction of allenes requires precise control over regio- and stereoselectivity. Intriguingly, the authors have addressed this issue by introducing a bifunctional cyano group into the allene substrate. This modification enables high selectivity in the transformation, facilitating the formation of three consecutive C-C bonds and leading to the synthesis of multisubstituted benzotropones. The subsequent exploration of substrate scope demonstrates excellent tolerance towards various functional groups, highlighting the broad applicability of this protocol. The synthetic applications showcased in the study further emphasize the potential utility of this methodology. To elucidate the mechanism, the authors have conducted a series of mechanistic studies. These investigations shed light on the roles of the nickel catalyst, organozinc reagent, and key intermediate, contributing to a clearer understanding of the reaction pathway.

In summary, this work presented by Chen and their colleagues represents an important contribution to the field, showcasing a significant advancement in transition metal-catalyzed carbomatalation chemistry. The broad substrate scope and excellent selectivity observed in this study open up new avenues for the synthesis of complex and valuable molecules. In my opinion, the manuscript warrants acceptance for publication in *Nature Communications*, pending the authors' careful attention to the following suggestions raised.

1. I am wondering why the carbene ligand is the most efficient ligand than other N-based ligands that are broadly used in the nickel-catalyzed carbonylative reaction, please clarify it.
2. Regarding the organozinc reagent, the alkylated zinc reagent is necessary to participate in the following cyclization sequence. In this context, it would be pertinent to consider whether benzyl zinc halide or the simplest methyl zinc halide is suitable for this transformation.
3. The inclusion of an ortho-cyano group on the arene substrate plays a crucial role in preserving the desired reactivity. However, if alternative functional directing groups, such as ester groups, were employed instead, it would be interesting to explore the impact on the reaction.
4. Deuterium experiments are performed to exclude the possibility of the intramolecular metal shift. Control experiments reveal that organozinc reagent is vital for the intramolecular cyclization of intermediate 4a. Does additional organozinc reagent serve as the base to promote the cyclization process? If it is, please clarify it.
5. A very recent Ni-catalyzed three-component carbonylative reaction: *ACS Catal.* 2023, 13, 4111–4119 might be cited.
6. Several typos should be corrected, including: Page 7, line 9: t is missing in 1,3-diketone; Page 8, line 3: l is missing in acylzincation; Page 9, conclusion part; Page 8, Fig 6c.

Reviewer #2:

Remarks to the Author:

Prof Chen and co-workers developed a new type of carbonylative reaction with CO gas under mild conditions. A NHC ligand promotes the reactions with moderate to good yields and excellent stereoselectivity. In addition, it's an efficient method for the synthesis of diverse substituted benzotropone derivatives by Nickel-based catalysts. In this strategy, one-pot three-component tandem acylzincation/cyclization sequence of allene and alkylzinc reagent with 1 atm of CO were achieved, and

three carbon-carbon bonds were constructed in one step. However, This manuscript could be accepted by Nature Communication after minor revision.

Other comments,

(1) In Fig 6 a), the substrate 1n was synthesized to verify the effect of the cyano group, and intermediate 4n shown Z/E value. The Z/E value of product 4a should be exhibited in Fig 2.

(2) In the final of this manuscript, there exists a conceptual mistake, and it must be revised. In sentence "..... and δ - lactam with good yield and enantioselectivity, which can be..." , your target molecules are racemic, do not have enantioselectivity ratio (er) value. "enantioselectivity" must be modified.

(3) In the proposed mechanism, D was proposed to be the key intermediate, followed by protonation and intramolecular condensation to give the final product. Where does the proton come from, especially in your basic conditions?

The detailed response to the reviewers' comments:

Reviewer #1 (Remarks to the Author):

Chen and coworkers have made an exciting advancement in the realm of transition metal-catalyzed carbometallation chemistry. Specifically, they have successfully achieved the nickel-catalyzed acylzincation of allenes using organozincs and CO, providing an efficient method for constructing fully substituted benzotropones, which are commonly found in various bioactive natural products. The same research group previously reported on the seminal work to probe the feasibility of acylmetallation of activated unsaturated hydrocarbons earlier this year (Nat. Synth. 2023, 2, 261). Comparing with the reported polarized unsaturated hydrocarbons such as ynamide and enones by the same group, the allene functional group presents a unique challenge due to its three potential reaction sites. Achieving the acylmetallation reaction of allenes requires precise control over regio- and stereoselectivity. Intriguingly, the authors have addressed this issue by introducing a bifunctional cyano group into the allene substrate. This modification enables high selectivity in the transformation, facilitating the formation of three consecutive C-C bonds and leading to the synthesis of multisubstituted benzotropones. The subsequent exploration of substrate scope demonstrates excellent tolerance towards various functional groups, highlighting the broad applicability of this protocol. The synthetic applications showcased in the study further emphasize the potential utility of this methodology. To elucidate the mechanism, the authors have conducted a series of mechanistic studies. These investigations shed light on the roles of the nickel catalyst, organozinc reagent, and key intermediate, contributing to a clearer understanding of the reaction pathway.

In summary, this work presented by Chen and their colleagues represents an important contribution to the field, showcasing a significant advancement in transition metal-catalyzed carbometallation chemistry. The broad substrate scope and excellent selectivity observed in this study open up new avenues for the synthesis of complex and valuable molecules. In my opinion, the manuscript warrants acceptance for publication in Nature Communications, pending the authors' careful attention to the following suggestions raised.

Response: Thanks for reviewer 1's positive comments.

1. I am wondering why the carbene ligand is the most efficient ligand than other N-based ligands that are broadly used in the nickel-catalyzed carbonylative reaction, please clarify it.

Response: Thanks for reviewer 1's question. We envision that the electron-rich character of NHC ligand would make the nickel metal center more electronically rich, thus facilitating the migratory insertion of acylnickel species towards allene moiety, while the strong coordination with the nickel also could prevent the undesired CO coordination to deactivate the catalyst reactivity.

In the revised manuscript, we added following comments: "while electron-rich NHC ligand L3 could make the nickel center more electronically rich, thus accelerate the migratory insertion and improve the reaction efficiency, affording the product 3a in 57% yield (entry 5).⁶⁸⁻⁷⁰"

2. Regarding the organozinc reagent, the alkylated zinc reagent is necessary to participate in the following cyclization sequence. In this context, it would be pertinent to consider whether benzyl zinc halide or the simplest methyl zinc halide is suitable for this transformation.

Response: We have examined these two substrates when investigating the substrate scope of this protocol, unfortunately, both of them failed to afford the corresponding products. With regard to methyl zinc chloride, the desired product was not obtained with 24% recovery of starting material. While most of the starting material recovered and there's no obvious newly-formed compound was found with the utilization of benzylic zinc reagent.

In the revised manuscript, we added these information: "The protocol was not suitable with the utilization of methyl and benzyl zinc nucleophiles".

3. The inclusion of an ortho-cyano group on the arene substrate plays a crucial role in preserving the desired reactivity. However, if alternative functional directing groups, such as ester groups, were employed instead, it would be interesting to explore the impact on the reaction.

Response: Thanks for reviewer 1's insightful suggestion. The allene substrate bearing an *ortho*-methyl ester group on the aromatic ring has been investigated, however, only the acylmetallation product was found (25% yield) and no desired benzotropone was obtained. Combining with the results shown in the original submission, these data collectively indicates that the *ortho* cyano group plays as a bifunctional role in this cascade sequence.

4. Deuterium experiments are performed to exclude the possibility of the intramolecular metal shift. Control experiments reveal that organozinc reagent is vital for the intramolecular cyclization of intermediate 4a. Does additional organozinc reagent serve as the base to promote the cyclization process? If it is, please clarify it.

Response: Thanks for reviewer 1's question. To verify the role of additional organozinc reagent during the cyclization step, we have performed some control experiments. When intermediate **4a** was subjected to the standard conditions without the addition of organozinc reagent, no cyclization product was observed. While the cyclization process could proceed smoothly to yield the final product in the presence of organozinc reagent even with the omission of catalyst and ligand. The above results indicate that the additional organozinc reagent probably serves as the base to promote the cyclization process. It should be also noted that the zinc amide species generated from the intramolecular cyclization in the real reaction mixture, may also participate the deprotonation in the intermolecular way, because the alpha C-H of enone intermediate is very acidic. Therefore, we cannot make very clear conclusion that the cascade cyclization was solely promoted by the excess zinc reagent.

5. A very recent Ni-catalyzed three-component carbonylative reaction: ACS Catal. 2023, 13, 4111–4119 might be cited.

Response: Thanks for reviewer 1's suggestion. It has been cited as ref. 44.

6. Several typos should be corrected, including: Page 7, line 9: t is missing in 1,3-diketone; Page 8, line 3: l is missing in acylzincation; Page 9, conclusion part; Page 8, Fig 6c.

Response: Thanks for reviewer 1's suggestion. All the above-mentioned typos have been corrected.

Reviewer #2 (Remarks to the Author):

Prof Chen and co-workers developed a new type of carbonylative reaction with CO gas under mild conditions. A NHC ligand promotes the reactions with moderate to good yields and excellent stereoselectivity. In addition, it's an efficient method for the synthesis of diverse substituted benzotropone derivatives by Nickel-based catalysts. In this strategy, one-pot three-component tandem acylzincation/cyclization sequence of allene and alkylzinc reagent with 1 atm of CO were achieved, and three carbon-carbon bonds were constructed in one step. However, this manuscript could be accepted by Nature Communication after minor revision.

Response: Thanks for reviewer 2's positive comments.

Other comments:

(1) In Fig 6 a), the substrate **1n** was synthesized to verify the effect of the cyano group, and intermediate **4n** shown *Z/E* value. The *Z/E* value of product **4a** should be exhibited in Fig 2.

Response: Thanks for reviewer 2's suggestion. The *Z/E* ratio of intermediate **4a** that we have isolated in one of the screening conditions is around 15:1, however, the amount of intermediate **4a** in most cases is too small to determine the exact *Z/E* ratio. Only 2-7% **4a** was observed in condition of entries 2-11 in Fig. 2, which has exceeded the test limits on ¹H NMR analysis.

(2) In the final of this manuscript, there exists a conceptual mistake, and it must be revised. In sentence "... and δ -lactam with good yield and enantioselectivity, which can be..." , your target

molecules are racemic, do not have enantioselectivity ratio (er) value. “enantioselectivity” must be modified.

Response: Thanks for reviewer 2’s comment. It has been corrected.

(3) In the proposed mechanism, D was proposed to be the key intermediate, followed by protonation and intramolecular condensation to give the final product. Where does the proton come from, especially in your basic conditions?

Response: Thanks for reviewer 2’s insightful question. We have carried out some deuterium experiments to rule out the possibility of the intramolecular metal shift (please see the Supplementary Information for details), and the zinc intermediate was trapped in the reaction mixture prior to the aqueous work-up procedure. We envisioned that the basic zinc intermediate might be trapped with solvent DMA, unfortunately, we cannot afford the anhydrous DMA- d_9 as solvent.

Reviewers' Comments:

Reviewer #1:

Remarks to the Author:

All of my concerns have been addressed.

Reviewer #2:

Remarks to the Author:

The questions have been taken care of. However, this manuscript could be accepted by Nature Communication.